# Comparison of Topographic Roughness of Layered Deposits on Mars

**Wei Cao** [1], **Zhiyong Xiao** [1,2,*], **Fanglu Luo** [1], **Yizhen Ma** [1] **and Rui Xu** [1]

[1] Planetary Environmental and Astrobiological Research Laboratory, School of Atmospheric Sciences, Sun Yat-Sen University, Zhuhai 519000, China; caow9@mail.sysu.edu.cn (W.C.); luoflu@mail2.sysu.edu.cn (F.L.); mayzh28@mail2.sysu.edu.cn (Y.M.); xurui53@mail2.sysu.edu.cn (R.X.)
[2] Centre for Excellence in Comparative Planetology, Chinese Academy of Sciences, Hefei 230026, China
[*] Correspondence: xiaozhiyong@mail.sysu.edu.cn; Tel.: +86-181-0865-0403

**Abstract:** Impact craters with layered ejecta deposits are widespread on Mars. Prevailing views suggest that such ejecta were formed due to the involvement of target water and/or water ice in the impact excavation and/or the post-deposition movement of the impact ejecta. The long-runout landslides and lobate debris aprons that are likely formed due to the involvement of water ice are used as analogs to compare roughness at multiple scales, considering that these three landforms share some similarities in their geomorphology. Analog studies of the morphological similarities and differences of layered ejecta deposits with different emplacement mechanisms are an important approach to untangling how layered ejecta deposits might form on Mars and beyond. Earlier morphological comparisons were usually based on qualitative descriptions or one-dimensional topographic roughness characteristics at given azimuths; however, the emplacement processes of layered deposits are recorded in two-dimensional topography and at multiple scales. In this study, we designed a multiwavelet algorithm to characterize the multi-scale topographic roughness of different forms of Martian layered deposits. Our comparisons show that the inner facies of the layered ejecta deposits and long-runout landslides exhibited similar roughness characteristics, and the outer facies of the layered ejecta deposits were more similar in roughness to lobate debris aprons. This study highlights the importance of the spatial resolution of digital terrain models in characterizing fine topographic fluctuations on layered ejecta deposits, providing additional insights into the possible emplacement mechanisms of different parts of layered ejecta deposits.

**Keywords:** Mars; impact crater; layered deposits; topography roughness

## 1. Introduction

On planetary surfaces, the continuous ejecta deposits of typical impact craters are emplaced by ballistically ejected fragments which exhibit a gradual decline in the surface elevation from the crater rim [1]. Most impact craters on Mars (~89% of global cataloged craters ≥5 km in diameter) [2] are commonly surrounded by one or more layers of ejecta deposits, where the ejecta blankets are composed of complete or partial sheets of fluidized materials (Figure 1a). These layered ejecta deposits (termed LEDs) show a large number of unique morphologic features, such as lobate aprons, elevated margins, radial grooves and ridges, and an absence of obvious secondary craters [3]. With distinctive morphologies, LEDs are also widespread on planetary bodies that may contain substantial water ice in the shallow target, such as the icy moons of the outer Solar System planets [4]. In general, impact craters with LEDs are typically regarded as a geomorphological indicator of the existence of subsurface liquid water or water ice [5,6].

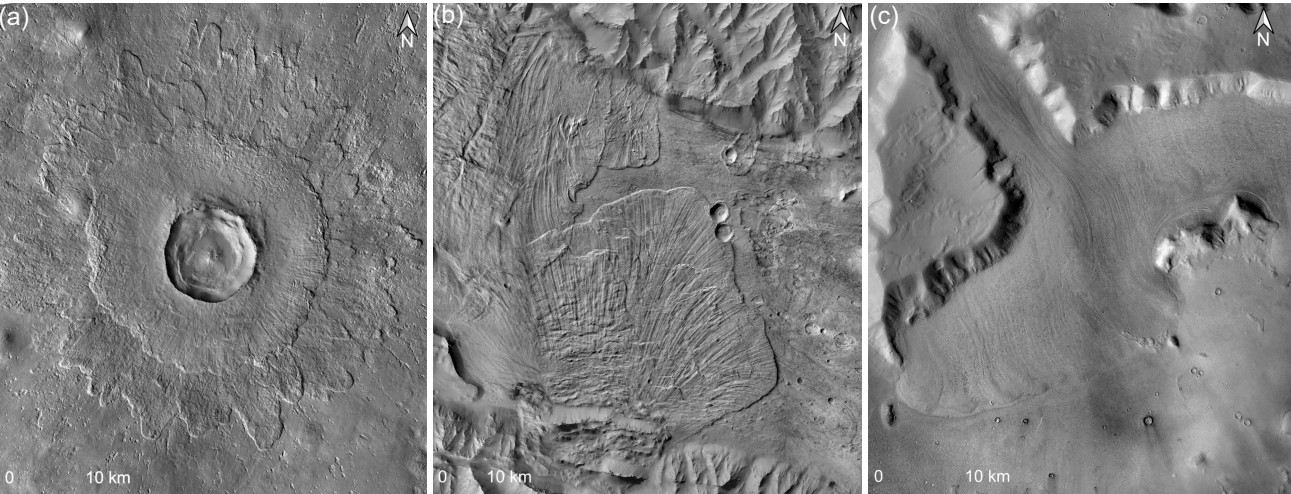

**Figure 1.** Various forms of layered deposits on Mars. (**a**) Layered ejecta deposits around the Steiheim crater (rim-to-rim diameter $D$ = 11.3 km; center coordinates: 54.6°N, 190.7°E). (**b**) Long-runout landslides (center coordinates: 11.0°S, 67.0°W) in the Valles Marineris. (**c**) A lobate debris apron (LDA) on Mars that is centered at 49.0°N, 22.0°E.

Understanding the emplacement mechanisms of Martian LEDs is critical to reveal whether water-ice materials (i.e., target volatiles) were involved in the impact cratering. Earlier comparative studies emphasized that the apparent similarities of various forms of layered deposits may be used as an indicator to understand the emplacement mechanisms of Martian LEDs [3,5,7–11]. Radial grooves, one of the flow-related features widespread on Martian LEDs, show unique morphologies and distributions that may influence the ejecta's emplacement process [5,12]. These radial grooves resemble longitudinal grooves formed on long-runout landslides (Figure 1b), which serve as analogs to radial grooves in Martian LEDs [9,13]. In addition, the presence of expanded secondary craters at the margins of layered ejecta deposits indicates the existence of water ice in the deposits [14]. The occurrence of concentric lineation in layered ejecta deposits, which appear similar to the concentric structures developed in the crater fills of LED craters, suggests that LEDs (e.g., double layered ejecta; DLE) may form via a post-impact ice deposition [14,15]. Viscous flow features (VFFs), surface deposits with lobate aprons that may be formed due to the involvement of glacier ice [16], have a similar morphology to LEDs. Lobate debris aprons (LDAs) are a subtype of VFFs [17] that have a similar morphology to LEDs, such as the tongue-shaped flow fronts and interior linear structures (Figure 1c).

The topographic roughness of planetary landscapes contains important information about their origin. Topographic roughness on horizontal scales of millimeters to kilometers is frequently used to analyze the apparent similarities between studied topographies and their analogs [18–21]. Earlier works analyzed the topographic roughness of LEDs and their analogs based on qualitative comparisons, indicating diverse possible mechanisms for LEDs on Mars [3,9,19]. Recently, quantitative comparisons of the apparent similarities of LEDs focused on topographic information in one dimension. For example, Pietrek et al. [13] found that the radial topographic profiles of LEDs and long-runout landslides feature similar fractal parameters, indicating a similar mode of emplacement. However, the selected one-dimensional topographic profiles were designed for directional features, while the entire LEDs feature topographic variations at different azimuths and scales. Therefore, the longitudinal and perpendicular topography of LEDs features diverging fractal parameters when characterizing the same two-dimensional terrains, which may exhibit weak anisotropy (Figures 4 and 5 in [13]). In this study, we performed a thorough characterization of the morphological characteristics of various forms of layered deposits, which included features of various dimensions and at different azimuths.

Because topography could be considered as a time series, Fourier and wavelet methods have been widely used to characterize topography using unique functions and mother wavelets and their appropriate coefficients [22,23]. However, Fourier series and traditional wavelets (e.g., Haar wavelet) are sensitive to their periodic functions and piecewise content functions, which may result in inaccurate estimations of the topographic information, such as jump discontinuities in the signals derived from the Fourier series [24]. In comparison to these series and wavelets, the multiwavelet method is capable of encompassing topographical information on planetary terranes at multiple scales [25], and the results offer nondirectional topographic roughness parameters rather than the conventional roughness parameters [26] so that more topographic details can be obtained.

In this study, we updated this method and invented an algorithm to analyze the topographic roughness of various forms of layered deposits on Mars (Section 2). We found that well-preserved long-runout landslides and LDAs on Mars have distinctive topographic roughness characteristics (Section 3). Comparisons with the topographic roughness of LEDs around Martian craters provided new insights into their apparent similarities (Section 4).

## 2. Data and Method

### 2.1. Data

This study selected well-preserved long-runout landslides, LDAs, and LEDs of impact craters on Mars to investigate their topographic roughness (Table 1). High-resolution imagery and topographic data of Mars were used in the present investigation. Images obtained by the Context Camera (CTX [27]) onboard the Mars Reconnaissance Orbit mission have pixel scales of ~6 m, and they were used to investigate the morphologies of various layered deposits. We followed the standard processing pipeline for CTX frames using the USGS Integrated Software version 7.0.0 for Imagers and Spectrometers (ISIS; https://github.com/USGS-Astrogeology/ISIS3 (accessed on 16 April 2022)). Table A1 in Appendix A lists the IDs and available addresses of the CTX frames used in this study.

**Table 1.** Basic information on the various forms of layered deposits studied in this study.

| List | Names and Terrane Types | CTX Stereo-Pair Id * | Location |
|---|---|---|---|
| 1 | LED of Bacolor | CTX_006750_2133_007462_2133 | 33.1°N, 118.7°E |
| 2 | LED of Zunil | CTX_020211_1877_038250_1877 | 7.5°N, 166.1°E |
| 3 | LED of Tooting | CTX_002013_2035_002646_2036; CTX_016280_2037_016425_2037 | 23.1°N, 152.1°W |
| 4 | Landslide 1 | CTX_008181_1676_035384_1682 | 12.3°S, 69.5°W |
| 5 | Landslide 2 | CTX_045868_1722_046580_1715 | 7.9°S, 77.8°W |
| 6 | Lobate debris apron 1 (LDA 1) | CTX_020203_2178_026994_2170 | 40.0°N, 25.0°E |
| 7 | Lobate debris apron 2 (LDA 2) | CTX_008731_2218_024317_2201 | 40.2°N, 24.0°E |
| 8 | Lobate debris apron 3 (LDA 3) | CTX_008731_2218_024317_2201 | 39.8°N, 24.4°E |

* The data are open to the public and can be found at https://marssi.univ-lyon1.fr/ (accessed on 28 March 2022).

The studied long-runout landslides, LDAs, and LEDs of impact craters on Mars are usually dozens of meters thick and kilometers long in lateral extensions [17,28–30]. Widespread features developed in such layered deposits have diverse sizes (Figure 2), and using topographic data with comparable spatial resolutions, we compared the topographic roughness of various forms of layered deposits at the same scale. We created Digital Elevation Models (DEMs) from CTX stereo-paired images to obtain topographic information on targeted cases (Table 1). For this purpose, the DEM processing pipeline established by the MarsSI platform [31,32] was employed to reduce the effects of artifact patterns and invalid pixel patches using adaptable thresholds for stereo matching and correlation by Mars Orbiter Laser Altimeter (MOLA) DEMs sampled at 453 m/pixel [33]. The generated DEMs had a pixel resolution of 12 m/pixel and a vertical error of ~2.2 m [34].

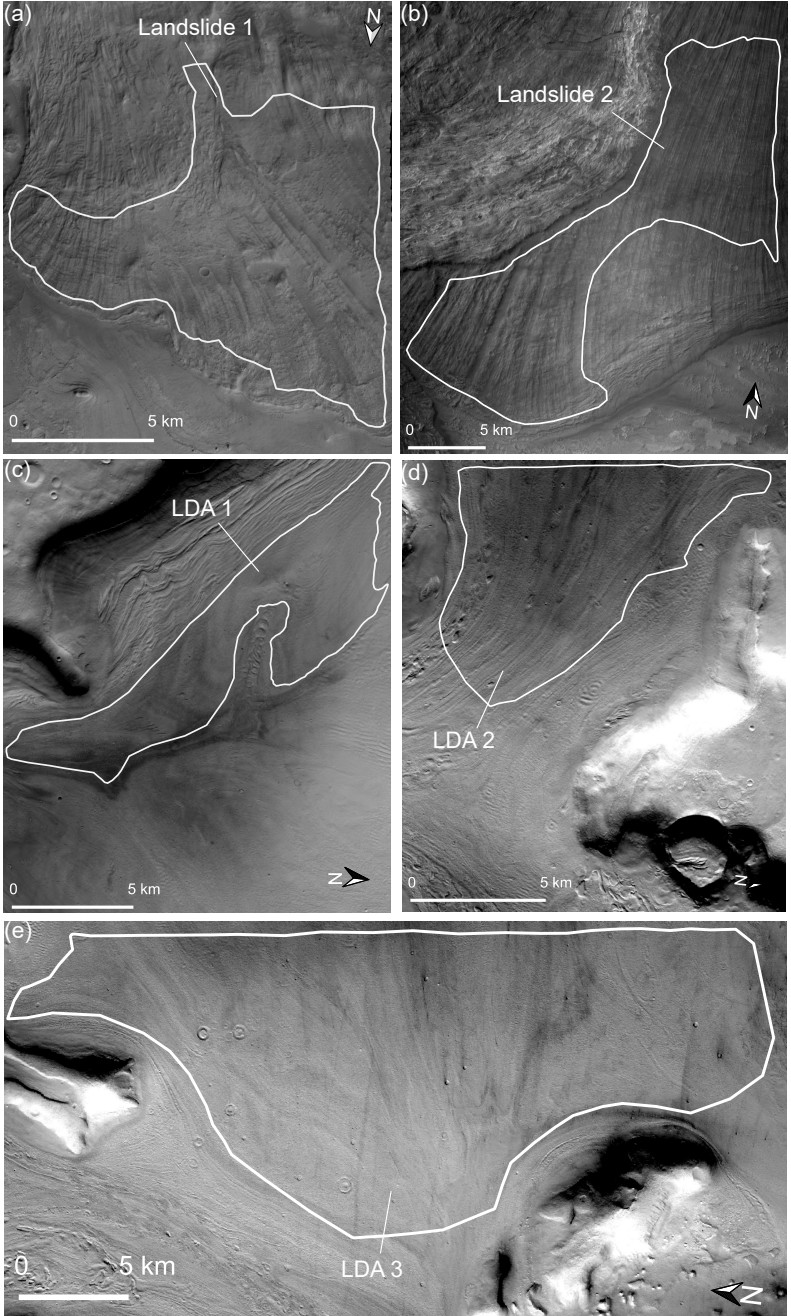

**Figure 2.** Morphology of the studied long-runout landslides and lobe debris aprons (LDAs) on Mars. The studied long run-out landslides are presented in (**a**,**b**). The studied LDAs are shown in (**c**–**e**). The resolutions of these CTX images are nearly 6 m/pixel (Table A1). All the locations of these studied layered deposits are provided and annotated in lists 4–8 in Table 1.

### 2.2. Method

Scale-dependent roughness parameters, such as root mean square (RMS) slope [18], and median differential slope (MDS) [35], are frequently used to investigate the topographic roughness of planetary surfaces. The given scales, i.e., the sizes of local windows for roughness calculation [36], have been measured based on the distinctive scientific requirements of the studies (e.g., 1/10 length of a profile [18]). Kreslavsky et al. [37] suggested that roughness measurements should characterize topography at well-defined scales. In this study, we referred to the reported dimensions of key features developed in layered deposits, such as radial grooves (~100–300 m [9]) and lobate fluidized deposits (≥1 km [38]), as

references for the determination of window sizes. The sizes of the calculation windows were restrained to 60 m (5 pixels; ~1/2 of 100 m), 132 m (11 pixels; ~1/8 of 1 km), and 252 m (21 pixels; ~1/4 of 1 km) as a trial to reduce the blocky appearance of roughness caused by large windows [36]. This scale selection accorded with the criteria that the scales for wavelet analysis must differ by factors of 2 [22,39].

The multiwavelet method (called V-system [25]) characterizes topographic information using its unique wavelet transform. The mother wavelet of the V-system is composed of Legendre's polynomials at various k degrees (k = 0, 1, 2, 3); the highest-degree (k = 3) multiwavelet (termed V3-system) can provide a greater approximation of complex signals than other multiwavelets [26] and it is applied for wavelet transform in this study.

Topographic roughness was quantified by the wavelet transform of the detrended topography based on a reference surface [23]. To characterize the fine topographic features of layered deposits, this study did not perform detrending calculations to remove the background topography [26], but we extracted the elevation differences between the central elevation and its surrounding elevations (termed as *T*) and they were used to characterize two-dimensional undulations in the topography of a specific window. The generalized wavelet transform can be expressed as follows:

$$F = \left| O \times T \times O^{\mathrm{T}} \right| \tag{1}$$

where *F* is the power spectrum including all the absolute values transformed from *T*, *O* is the V3-system matrix including the V3-system wavelets, and $O^{\mathrm{T}}$ is their transposed forms [25,26].

Surface roughness is defined as a coefficient of the variation (i.e., the ratio of standard deviation to the mean; termed CV) of *F* because CV can be applied as a variable to characterize varied topographic parameters (e.g., elevation, slope, etc. [40,41]). Higher roughness is related to higher spatial dispersion of the power spectrum. To distinguish it from the similar roughness method proposed by [26], the roughness method in this study was termed Improved V3-system Roughness (IV3R).

Although common statistical parameters (e.g., mean, median, etc.) are convenient to compare roughness, we suggest that the roughness value of each layered deposit should be considered in comparison, because they characterize multi-scale topographic information at their locations. Consequently, we established a comparison criterion based on the observation that long-runout landslides and LDAs show distinct morphologies that may create different roughness features (Figure 1b,c). Based on this theoretical difference, we calculated skewness to characterize the differences in the roughness of the studied layered deposits. In this study, higher skewness indicated a greater degree of roughness of the studied layered deposits. Because the mean values and standard deviation of each layered deposit were quite different and could not be used as a suitable measure of comparison, we considered that the mean values ($\mu$) and standard deviations ($\sigma$) of the roughness of these layered deposits could be used to differentiate roughness. To build a uniform criterion for calculating skewness, we calculated the mean of $\mu$ (termed $m\mu$) and $\sigma$ (termed $m\sigma$) of the studied long-runout landslides and LDAs. The calculation of skewness was expressed by

$$Skew = \frac{E(X - m\mu)^3}{m\sigma^3} \tag{2}$$

where *Skew* is the skewness of the roughness of the studied layered deposit. The roughness values of the layered deposit were annotated by *X*. *E* is the expected value of $(X - m\mu)^3$.

## 3. Results

### 3.1. Topography Roughness Represented by IV3R

As shown by the demonstration case of the fluidized deposits on the front of the LDA (centered at 39.4°N, 24.2°E; Figure 3a–d), the V3-system algorithm has a comparative advantage in deciphering detailed topographic characteristics in the layered deposits. We

calculated roughness using different methods (i.e., root mean square slope, RMS Slope; median different slope, MDS; Interquartile range of Curvature; IV3R) at the 132 m scale for comparison. All the studied roughness textures were displayed using the local cumulative cut–stretch function to obtain a clear observation of subtle topographic information. A comprehensive summary of the calculations of these compared methods is provided in Appendix A.

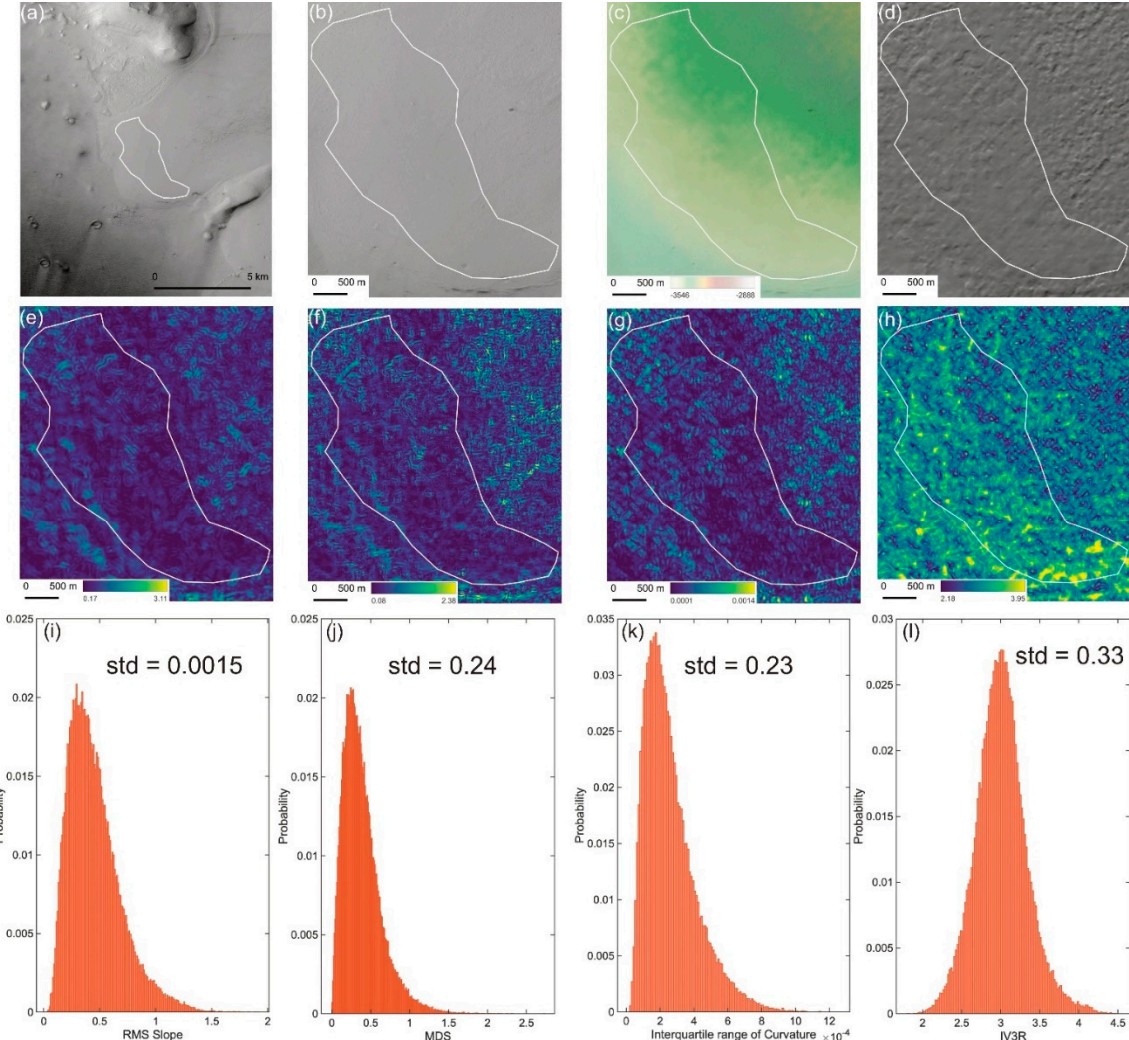

**Figure 3.** Fluidized deposits on the front of the lobate debris apron (LDA) outlined in white (center coordinates: 39.4°N, 24.2°E), used as a trial to demonstrate the advantages of IV3R. (**a**) Fluidized deposits on the front of the LDA show a flatter morphology compared to its surrounding terrains, and their inner topography, generated DEM, and shade relief of the DEM are shown in (**b**–**d**). The roughness parameters at the 132 m scale, including (**e**) RMS Slope, (**f**) MDS, (**g**) interquartile range of curvature, and (**h**) IV3R, are displayed using the local cumulative cut–stretch function. The histograms of these roughness parameters with bin widths are determined by the Freedman–Diaconis rule (implemented by the *histogram* function in MATLAB software version R2022b), and the standard deviations (stds) of the roughness values of the tested deposits are calculated in (**i**–**l**) to indicate the probability distributions of these roughness parameters.

To illustrate the advanced capabilities of the IV3R in characterizing the full-scale and all-azimuth roughness of layered deposits, we conducted a comparative analysis of the three forms of flow-like deposits, considering that these flow-like deposits have flat topographies with complex local variations. The results demonstrated that the IV3R could exhibit more details in the roughness compared to conventional approaches (Figure 3e–h).

Moreover, the histograms and standard deviations (stds) of the roughness values calculated by different methods statistically demonstrated these textural differences, with the IV3R showing more dispersed distributions of the roughness values and a higher std (Figure 3i–l). This result indicated that the IV3R is capable of extracting more subtle roughness textures compared to other traditional methods.

### 3.2. Comparison of Roughness of Lobate Debris Aprons and Long-Runout Landslides on Mars

An assessment was performed to verify the reliability of the proposed criterion by determining the skewness of the multi-scale roughness of the studied long-runout landslides and layered deposits (LDAs). Additionally, we calculated the skewness of the IV3R of these layered deposits to make comparisons and presented our findings in Figure 4. The investigated long-runout landslides showed complex morphologies that are characterized by the radial textures developed along dispersed radial directions (Figure 2a,b). In contrast, well-persevered LDAs show a smooth and flat morphology (Figure 2c–e). Based on these morphological differences, it is deduced that the topography of long-runout landslides is rougher than that of LDAs.

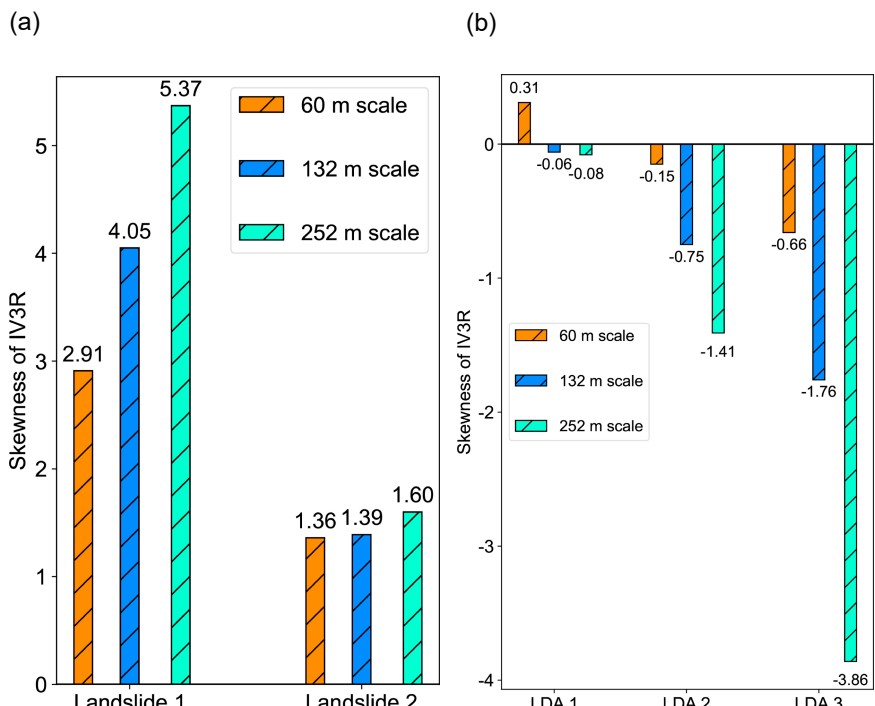

**Figure 4.** Comparison of topography roughness as represented by the skewness of IV3R for lobate debris aprons (LDAs) and long-runout landslides on Mars. (**a**) skewness values at multiple scales of the studied long-runout landslides. (**b**) skewness values at multiple scales of the studied LDAs. The 60 m scale and 132 m scale skewness values are highlighted by orange and blue stripes with diagonal textures, while a cyan diagonal stripe is used to represent the skewness at the 252 m scale. The numerical value of each skewness is shown on the top of these stripes.

By comparing the skewness of roughness, the disparity in morphologies between the studied long-runout landslides and LDAs was identified. Our results indicated that the skewness values of long-runout landslides were consistently greater than 0 and displayed an increasing trend as the scale increased (Figure 4a). Conversely, the skewness values of LDAs were below 0 and presented a decreasing trend as the scales increased (Figure 4b). In addition, the IV3R characteristics of Landslide 1 were associated with higher skewness than those of Landslide 2 at all scales. Our findings supported the suitability of the skewness of the IV3R as an indicator in the comparisons carried out in this study.

### 3.3. Roughness of Layered Ejecta Deposits of Martian Craters

To investigate the roughness properties of complete LEDs, we focused on the fresh LED crater called Bacolor, a young LED crater frequently studied in past research [3,7,42], to use as a suitable prototype (Table 1 and Figure 5a). While earlier studies have classified the Bacolor crater as a double-layered ejecta crater [12,28], an irregular morphology composed of small flow-related lobes and radial grooves are frequently visible on Bacolor LEDs (Figure 5b) and they may suggest distinct roughness properties. Therefore, we investigated ejecta deposits based on their different morphologies, visible boundaries, and similar spreading directions, as shown in the CTX optical images and generated DEMs (i.e., L1–L4 in Figure 5a). We focused on the lobate deposits on the southern parts of the Bacolor LEDs because they contained more distinctive, multiple boundaries of LEDs compared to those on the northern parts. The selected lobate deposits were constrained by their similar spreading directions (L1–L4).

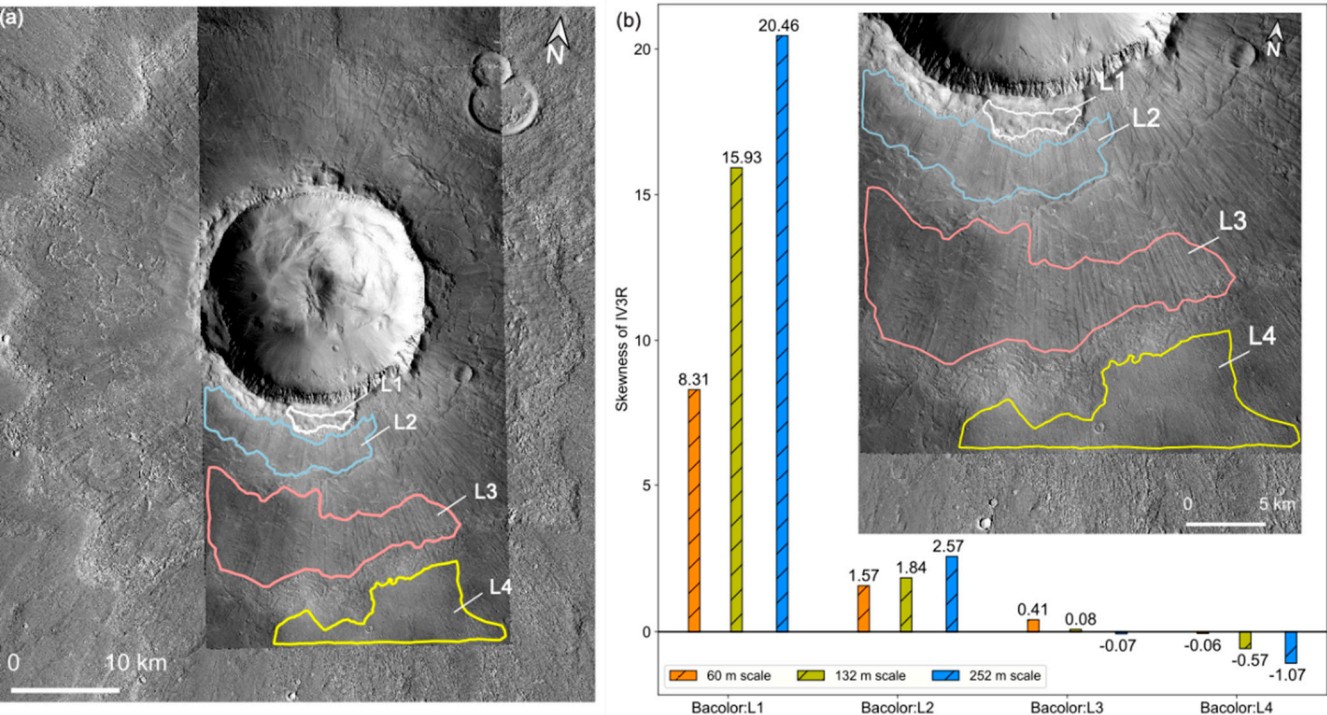

**Figure 5.** Topographic roughness of layered ejecta deposits of the Bacolor crater (center coordinates: 33.1°N, 118.7°E). (**a**) The topography of Bacolor crater. The studied LEDs are outlined in varied colors (white for L1, blue for L2, red for L3, and yellow for L4). The stereo-paired aligned image used for DEM generation (CTX_006750_2133_007462_2133) is displayed in black. The base image is derived from a seamless mosaic of CTX images P17_007752_2140_XN_34N242W, P18_008108_2126_XN_32N241W, P21_009044_2132_XN_33N241W, and P22_009677_2133_XN_33N241W. (**b**) Skewness distribution of IV3R at multiple scales. Details of the distinctive morphologies of the studied LEDs are screened using a small window in the upper right corner.

Our investigation revealed that Bacolor LEDs possess heterogeneous roughness properties as evidenced by the various skewness values at different scales. The inner-most facies (L1), located in the vicinity of the rim, manifested a rough topography formed by post-ejecta depositions and was characterized by higher skewness values at multiple scales (8.31–20.46) compared to L2–L4. The inner facies L2, located within 0–2 radii from the parent crater rim, had smaller skewness values (1.57–2.57) compared to L1, which may be attributed to its morphology, as it is formed by radial grooves and lobate deposits on the rough surface. In contrast, the outer facies (L3) and the outer-most facies (L4) exhibited diverse roughness characteristics at multiple scales, with the skewness of their roughness

tending to decrease with increasing scales. The skewness values of L3 and L4 were found to be approximately zero or below this threshold.

## 4. Discussion

### 4.1. A Possible Connection between Topographic Roughness and Emplacement Process of Layered Deposits on Mars

Landslides are downward and outward movements of slope-forming materials, which are usually composed of rocks, soils, etc. [43]. An ongoing subject of debate concerning the emplacement mechanisms of long-runout landslides on Mars is whether dry granular flows or ice-rich materials contribute to the reduction in the basal surface [44,45]. It has been demonstrated that the landslides, dry and/or ice-rich materials on Mars could both travel with high velocities (from 10 to 100 m/s) on the basal surface and form long-runout lobes, forming long-runout landslides [45–47]. In contrast, LDAs have been known to result from the creeping motion caused by the slow deformation of ice–rock mixtures influenced by the slope gradient [48–50]. In this study, the comparison of the skewness of long-runout landslides and LDAs indicated that long-runout landslides exhibit varying values of skewness, along with an increasing trend at multiple scales (Figure 4a,b). This difference may be connected to the diverse emplacement mechanisms that are specific to these layered deposits.

Based on the roughness differences between long-runout slides and LDAs, it has been proposed that LEDs exhibiting roughness characteristics similar to long-runout landslides or LDAs may indicate diverse formation processes. Utilizing this rationale, we compared the skewness of the roughness of the Bacolor LEDs to that of long-runout landslides, as well as LDAs (Figure 4), revealing diverse roughness characteristics among LEDs, long-runout landslides, and LDAs (L2–L4 in Figure 5). Specifically, L2 displayed skewness values and an increasing trend like those seen in long-runout landslides. On the contrary, L3 and L4 indicated skewness values and trends close to those of LDAs. Furthermore, L3 displayed a higher skewness at multiple scales compared to L4. This skewness difference was consistent with the fact that L3 showed a rougher topography than L4. The roughness comparisons suggest that LEDs are likely emplaced by a combination of different processes that are partly similar to those of slumps, long-runout landslides, and LDAs.

### 4.2. Heterogeneous Roughness for Layered Ejecta Deposits on Mars

It has been observed that the special LEDs of impact craters covering partial continuous ejecta blankets are widespread on the Martian surface (i.e., classified as Hu/Sm BL/SL/Am/Sm-layered deposit types, ~43% of 44,663 LED craters [28,29]). These LEDs have diverse morphologies and are located at different azimuths on the ejecta blankets of Martian impact craters (e.g., multiple layered ejecta [51]), which may indicate diverse emplacement mechanisms. Consequently, we suggest that the observed large heterogeneity in the roughness characteristics of such LEDs may indicate that the emplacement mechanisms of LEDs are highly diverse at different azimuths and/or mobility ranges.

In this study, we used Zunil (Figure 6a) as a prototype due to its visible lobate ejecta layers and the incomplete coverage of the continuous ejecta blankets at various azimuths [52]. We also investigated the Tooting crater, whose continuous ejecta blanket is composed of multiple lobate-layered ejecta (Figure 7a). Moreover, the Tooting crater is one of the best-preserved examples for studying LEDs on Mars [53]. According to the definition provided by Barlow [51], lobate-layered ejecta covering partial continuous ejecta blankets are termed partial continuous deposits. In agreement with this definition, the lobate-layered ejecta found in the Zunil and Tooting craters were referred to as partial continuous LEDs in this study.

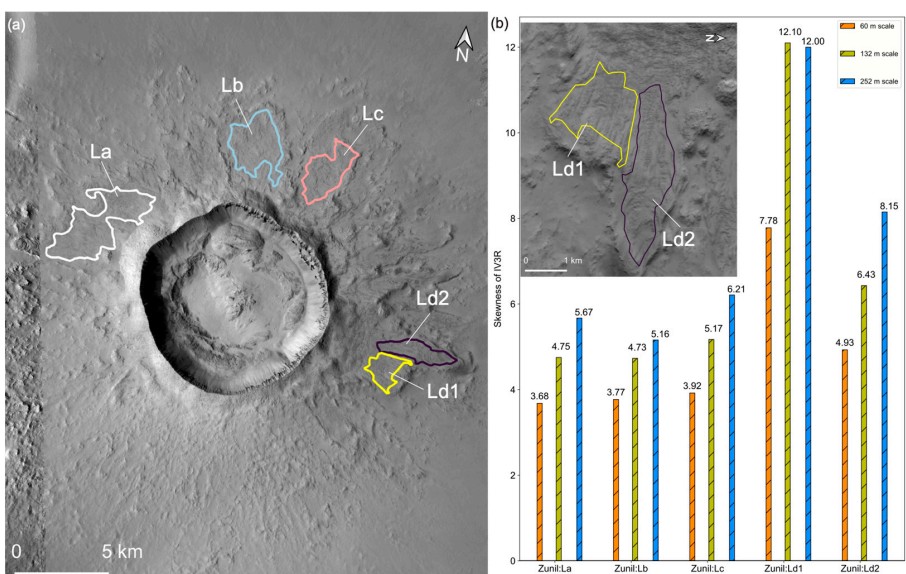

**Figure 6.** The topography and multi-scale skewness of the chosen LEDs on the facies of the Zunil crater (center coordinates: 7.5°N, 166.1°E). (**a**) The studied LEDs are outlined in a variety of colors. The brighter base image is the stereo-paired aligned image (CTX_020211_1877_038250_1877). The base image is a seamless mosaic of CTX images G05_020211_1877_XN_07N193W and G06_020554_1879_XI_07N194W. (**b**) Multi-scale skewness of IV3R. Detailed topography of Ld1 and Ld2 is screened using a small window in the upper left corner, and its base image is the stereo-paired aligned image CTX_020211_1877_038250_1877.

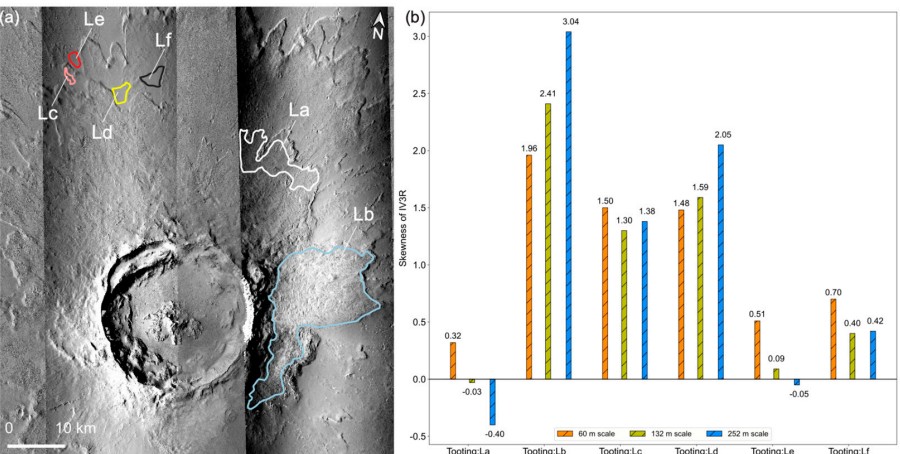

**Figure 7.** The morphology of the studied LEDs on the facies of the Tooting crater (center coordinates: 23.1°N, 152.1°W) and their multi-scale skewness. (**a**) The topography of the Tooting crater and its studied deposits, outlined in a variety of colors. Darker images are the CTX stereo-paired images CTX_002013_2035_002646_2036 and CTX_016280_2037_016425_2037. The base image is a seamless mosaic of CTX images K01_053927_2040_XI_24N151W, D03_028583_2035_XI_23N152W, B17_016412_2036_XN_23N152W, and B17_016280_2037_XN_23N151W. (**b**) Multi-scale skewness of IV3R.

In this study, we investigated the roughness characteristics of the partial continuous LEDs of the Zunil crater, which exhibited heterogeneity at multiple scales that may be related to the varying formation processes of the layered deposits. We focused on the northernmost area of the LEDs on the Zunil crater, where three partial continuous LEDs were extracted for comparison due to their similar locations (~0–2 radii from the parent crater rim; La–Lc; Figure 6a) and morphologies (Figure A1). These LEDs exhibited similarities in their skewness values (3.68–6.21; Figure 6b) and trends compared to those observed in

long-runout landslides (Figure 4a). In addition, we considered a unique partial continuous LED with diverse fluidized deposits (Ld1 and Ld2) on this LED (Figure 6b). Our analysis revealed that the skewness of Ld1 exhibited a sudden increase from 7.78 to 12.10 at the 60 m and 132 m scales, followed by a slight decrease from 12.10 to 12.00 at the 132 m and 252 m scales. Ld2 exhibited similar skewness distributions (with values ranging from 4.93 to 8.15 with an increasing trend) to those observed in La–Lc.

Similar results were also found in the comparison of the roughness of partial continuous LEDs on the Tooting crater. Specifically, the La–Lb LEDs were located on the continuous ejecta blankets but exhibited diverse morphologies (Figure 7a). By comparing the various ranges of elevation, it was discovered that the smooth topography of La resembled the layered deposits of the outer facies (at elevations varying from nearly −3905 to −3834 m; Figure 8a,b). Consequently, we considered that Lb was the partial continuous LED on the inner facies of the Tooting crater, while La was the partial continuous LED on the outer facies. The investigation at multiple scales of the skewness of these partial continuous LEDs revealed that Lb exhibited higher skewness values (1.96–3.04) and an increasing trend, whereas La exhibited lower skewness values (−0.4–0.32) and a decreasing trend (Figure 7b). Furthermore, compared to the skewness of long-runout landslides and LDAs (Figure 4), La exhibited similar skewness values and trends to those of LDA 1 (−0.08–0.31), and Lb exhibited similar values and changing trends of skewness to those of Landslide 1 (2.91–5.37).

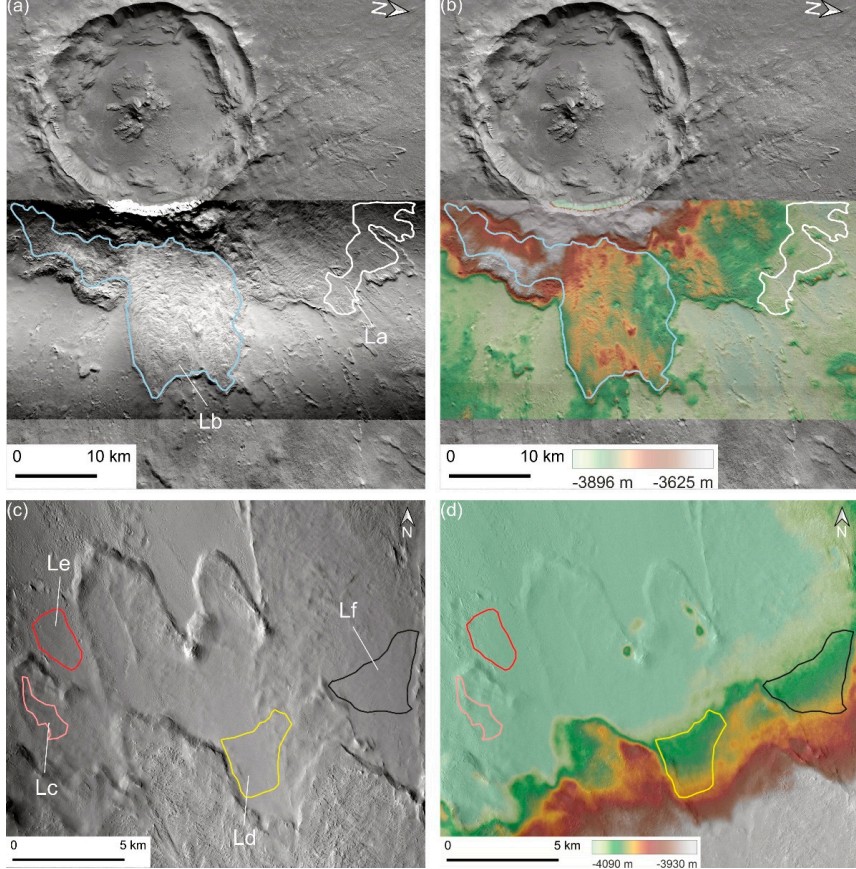

**Figure 8.** Topography and the generated DEMs of the studied deposits on the Tooting LEDs that are shown in Figure 7. The studied LEDs La and Lb (center coordinates: 23.9°N, 152.5°W) are presented in the stereo-paired image CTX_016280_2037_016425_2037 and its DEMs are shown in (**a,b**), and those of Lc–Lf (center coordinates: 23.5°N, 151.8°W) present in the stereo-paired image CTX_002013_2035_002646_2036 and its DEM are displayed in (**c,d**). The base image in (**a,b**) is a seamless mosaic of CTX images K01_053927_2040_XI_24N151W, D03_028583_2035_XI_23N152W, B17_016412_2036_XN_23N152W, and B17_016280_2037_XN_23N151W.

It seems that the roughness similarities observed in the studied partial continuous LEDs (Figures 6–8) are connected to specific emplacement processes, such as those associated with long-runout landslides or LDAs. In contrast, heterogeneous roughness characteristics were found on the partial continuous LEDs of the Tooting craters. For example, some smooth, fluidized deposits on the partial continuous LEDs (Lc–Lf) exhibited comparable morphologies (Figure 8c,d), but had different roughness characteristics at multiple scales (Figure 7b). The skewness trends and values of Le were found to be similar to those observed for LDA 1 (Figure 4b), while Ld displayed skewness values that were comparable to those seen in Landslide 2 (Figure 4a). In addition, a unique skewness trend was observed for Lf, which displayed a slight increase in skewness on the scales from 132 to 252 m (from 1.30 to 1.38) immediately following a sudden decrease in skewness observed on the scales from 60 to 132 m (from 1.50 to 1.30). Although farther away than Lf, Lc exhibited diverse values (4.93–8.15) of skewness with differing trends across multiple scales (Figures 7b and 8c).

In summary, roughness comparisons revealed heterogeneous topographic characteristics, and the overall similarities of the quantified roughness of the various forms of layered deposits suggested that the inner facies of layered ejecta deposits are more like long-runout landslides, and the outer facies are more similar to lobate debris aprons.

## 5. Conclusions

In this study, we compared the roughness properties of LEDs on Mars by inventing a new multiwavelet roughness method. Layered deposits such as well-preserved LED craters, long-runout landslides, and LDAs were used as a demonstration trial. The proposed roughness captured the topographic variations of the layered deposits in diverse locations and at different azimuths on Mars. We conducted comparative studies of roughness characteristics at multiple scales to investigate the formation process of LEDs. The skewness of the roughness at multiple suitable scales indicated that there were morphological similarities between the studied layered ejecta deposits. The skewness values and trends at multiple scales were used as an indicator for quantifying the morphological similarities of layered deposits. The heterogeneous roughness of LEDs, compared to other layered deposits, showed that the inner facies of LEDs shared similar topographic variations to those resulting from landslides, while the outer facies coincided with those produced by lobate debris aprons. These results may provide a new constraint in the future investigation of the formation mechanisms of LEDs.

**Author Contributions:** Conceptualization, Z.X.; methodology, W.C. and Z.X.; software, W.C.; validation, Z.X. and W.C.; formal analysis, Z.X. and W.C.; investigation, Z.X., W.C., F.L., Y.M. and R.X.; resources, Z.X. and W.C.; data curation, W.C.; writing—original draft preparation, Z.X. and W.C.; writing—review and editing, Z.X. and W.C.; visualization, W.C.; supervision, Z.X.; project administration, Z.X.; funding acquisition, Z.X. and W.C. All authors have read and agreed to the published version of the manuscript.

**Funding:** The authors are supported by the B-type Strategic Priority Program of the Chinese Academy of Sciences (grant XDB41000000), the fellowship of the China Postdoctoral Science Foundation (2022M723575), the pre-research Project on Civil Aerospace Technologies (D020201, D020202), and the China Manned Space Engineering Program.

**Data Availability Statement:** All the base data (roughness maps and MATLAB codes) used in this study are uploaded to the data repository in the public domain (available at https://doi.org/10.5281/zenodo.7743820 accessed on 3 January 2023).

**Conflicts of Interest:** The authors declare no conflict of interest.

## Appendix A

The Reproduction of roughness methods for comparison (RMS Slope, MDS, and Interquartile range of Curvature) in the generated DEM from the MarsSI platform.

- Root Mean Square Slope (RMS Slope) and Median Different Slope (MDS)

The one-dimensional RMS Slope is reproduced based on Equations (5) and (6) in Shepard et al., (2001) [18], and the MDS is reproduced based on Equation (1) in Kreslavsky et al., (2000) [35]. The differences in the reproduced Slopes were that we utilized a 2-norm measure to characterize the two-dimensional slopes, which is expressed by

$$S = \arctan\sqrt{\left(\tan\alpha_x{}^2 + \tan\alpha_y{}^2\right)} \times 180/\pi \tag{A1}$$

where *S* denotes the reproduced slopes (deg.), $\tan\alpha_x$ is the one-dimensional slope on the *x*-axis (e.g., longitude), and $\tan\alpha_y$ is the one-dimensional slope on the *y*-axis (e.g., latitude).

- Interquartile range of Curvature (IQR-C)

Although Kreslavsky et al., (2013) [37] provided multi-scale roughness maps using this method and one-dimensional LOLA data, we referred to the equations in Section 3.2 of Kokhanov et al., (2019) [54]—which is an updated version of that found in Kreslavsky et al., (2013) [37]—for the one-dimensional IQR-C, which is considered to be more suitable for characterizing two-dimensional roughness on gridded DEMs. We calculated the 2-norm of the one-dimensional IQR-C on the *x*-axis and *y*-axis, respectively.

$$C = \sqrt{\left(C_x{}^2 + C_y{}^2\right)} \tag{A2}$$

where *C* is the two-dimensional IQR-C, $C_x$ is the one-dimensional IQR-C on the *x*-axis (e.g., longitude), and $C_y$ is the IQR-C on the *y*-axis (e.g., latitude).

The MATLAB procedure codes of these methods (including the codes of the IV3R) are available at https://doi.org/10.5281/zenodo.7743820 (accessed on 3 January 2023).

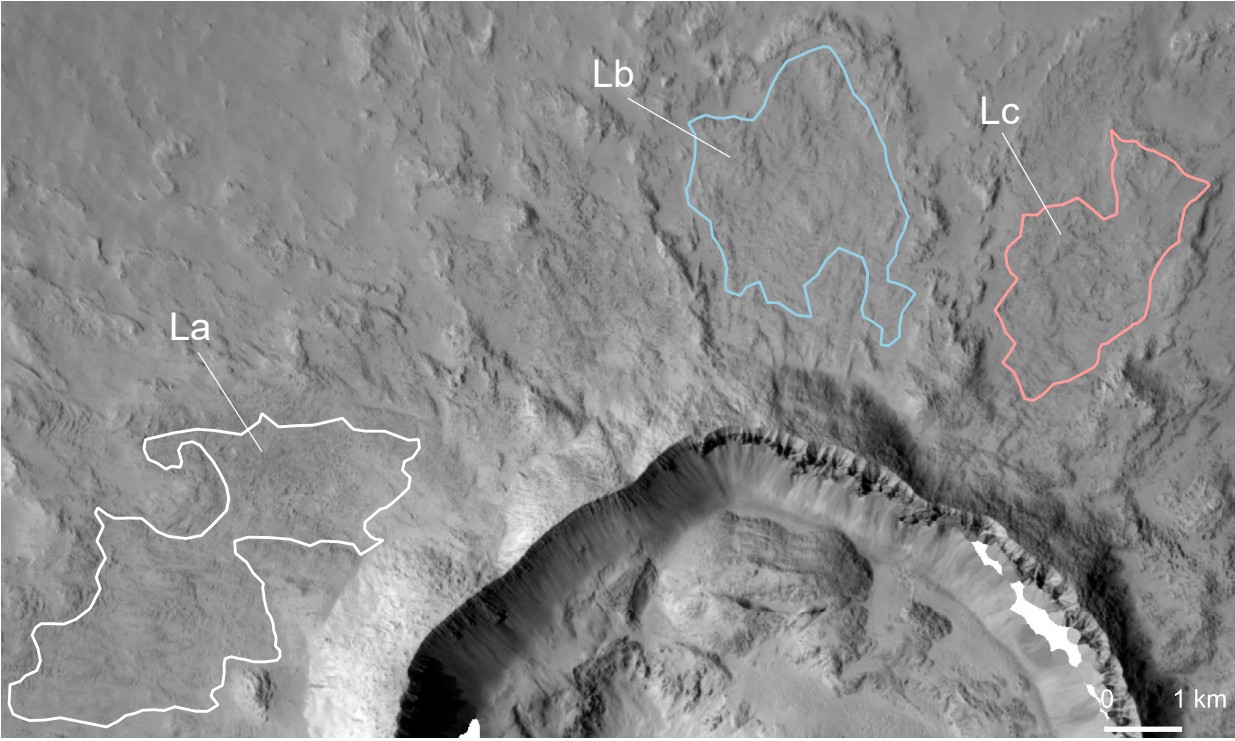

**Figure A1.** The detailed topography of three small lobes in the Zunil craters (La–Lc). All the LEDs are outlined in diverse colors. The base images are the stereo-paired aligned image (CTX_020211_1877_038250_1877) and a seamless mosaic of CTX images G05_020211_1877_XN_07N193W and G06_020554_1879_XI_07N194W.

**Table A1.** IDs and available addresses of CTX images used in this study.

| Figure | Data ID | Map Projection | Pixel Scale | Available Address |
|---|---|---|---|---|
| Figure 1a | G02_018944_2348_XI_54N168W | Orthographic | 6 m/pixel | https://ode.rsl.wustl.edu/mars/productPageAtlas.aspx?product_id=G02_018944_2348_XI_54N168W&product_idGeo=15222502 (accessed on 11 August 2011) |
| | G21_026302_2344_XN_54N169W | Orthographic | 6 m/pixel | https://ode.rsl.wustl.edu/mars/productPageAtlas.aspx?product_id=G21_026302_2344_XN_54N169W&product_idGeo=20807336 (accessed on 6 March 2012) |
| | G22_026724_2348_XN_54N169W | Orthographic | 6 m/pixel | https://ode.rsl.wustl.edu/mars/productPageAtlas.aspx?product_id=G22_026724_2348_XN_54N169W&product_idGeo=20807837 (accessed on 8 April 2012) |
| | G23_027014_2349_XN_54N168W | Orthographic | 6 m/pixel | https://ode.rsl.wustl.edu/mars/productPageAtlas.aspx?product_id=G23_027014_2349_XN_54N168W&product_idGeo=20808739 (accessed on 1 May 2012) |
| | P17_007736_2349_XI_54N169W | Orthographic | 6 m/pixel | https://ode.rsl.wustl.edu/mars/productPageAtlas.aspx?product_id=P17_007736_2349_XI_54N169W&product_idGeo=9088061 (accessed on 21 March 2008) |
| Figure 1b | B21_017688_1685_XN_11S067W | Orthographic | 5 m/pixel | https://ode.rsl.wustl.edu/mars/productPageAtlas.aspx?product_id=B21_017688_1685_XN_11S067W&product_idGeo=14562359 (accessed on 5 May 2010) |
| | P07_003632_1681_XI_11S068W | Orthographic | 5 m/pixel | https://ode.rsl.wustl.edu/mars/productPageAtlas.aspx?product_id=P07_003632_1681_XI_11S068W&product_idGeo=9042119 (accessed on 6 May 2007) |
| | P16_007113_1678_XN_12S067W | Orthographic | 5 m/pixel | https://ode.rsl.wustl.edu/mars/productPageAtlas.aspx?product_id=P16_007113_1678_XN_12S067W&product_idGeo=9081306 (accessed on 1 February 2008) |
| | P20_008906_1685_XN_11S067W | Orthographic | 5 m/pixel | https://ode.rsl.wustl.edu/mars/productPageAtlas.aspx?product_id=P20_008906_1685_XN_11S067W&product_idGeo=12280626 (accessed on 20 June 2008) |
| Figure 1c | P05_002890_2205_XI_40N337W | Orthographic | 6 m/pixel | https://ode.rsl.wustl.edu/mars/productPageAtlas.aspx?product_id=P05_002890_2205_XI_40N337W&product_idGeo=9038791 (accessed on 9 March 2007) |
| | P20_008942_2198_XN_39N337W | Orthographic | 6 m/pixel | https://ode.rsl.wustl.edu/mars/productPageAtlas.aspx?product_id=P20_008942_2198_XN_39N337W&product_idGeo=12280761 (accessed on 23 June 2008) |
| Figure 5 | P17_007752_2140_XN_34N242W | Orthographic | 6 m/pixel | https://ode.rsl.wustl.edu/mars/productPageAtlas.aspx?product_id=P17_007752_2140_XN_34N242W&product_idGeo=9088303 (accessed on 22 March 2008) |
| | P18_008108_2126_XN_32N241W | Orthographic | 6 m/pixel | https://ode.rsl.wustl.edu/mars/productPageAtlas.aspx?product_id=P18_008108_2126_XN_32N241W&product_idGeo=9091997 (accessed on 19 April 2008) |
| | P21_009044_2132_XN_33N241W | Orthographic | 6 m/pixel | https://ode.rsl.wustl.edu/mars/productPageAtlas.aspx?product_id=P21_009044_2132_XN_33N241W&product_idGeo=12281091 (accessed on 1 July 2008) |
| | P22_009677_2133_XN_33N241W | Orthographic | 6 m/pixel | https://ode.rsl.wustl.edu/mars/productPageAtlas.aspx?product_id=P22_009677_2133_XN_33N241W&product_idGeo=12283284 (accessed on 19 August 2008) |

**Table A1.** *Cont.*

| Figure | Data ID | Map Projection | Pixel Scale | Available Address |
|---|---|---|---|---|
| Figure 6a | G05_020211_1877_XN_07N193W | Orthographic | 7 m/pixel | https://ode.rsl.wustl.edu/mars/productPageAtlas.aspx?product_id=G05_020211_1877_XN_07N193W&product_idGeo=16377084 (accessed on 18 November 2010) |
| | G06_020554_1879_XI_07N194W | Orthographic | 6 m/pixel | https://ode.rsl.wustl.edu/mars/productPageAtlas.aspx?product_id=G06_020554_1879_XI_07N194W&product_idGeo=17469180 (accessed on 15 December 2010) |
| Figures 7a and 8 | B17_016280_2037_XN_23N151W | Orthographic | 6 m/pixel | https://ode.rsl.wustl.edu/mars/productPageAtlas.aspx?product_id=B17_016280_2037_XN_23N151W&product_idGeo=14247440 (accessed on 16 January 2010) |
| | B17_016412_2036_XN_23N152W | Orthographic | 7 m/pixel | https://ode.rsl.wustl.edu/mars/productPageAtlas.aspx?product_id=B17_016412_2036_XN_23N152W&product_idGeo=14249090 (accessed on 26 January 2010) |
| | D03_028583_2035_XI_23N152W | Orthographic | 6 m/pixel | https://ode.rsl.wustl.edu/mars/productPageAtlas.aspx?product_id=D03_028583_2035_XI_23N152W&product_idGeo=20497342 (accessed on 31 August 2012) |
| | K01_053927_2040_XI_24N151W | Orthographic | 6 m/pixel | https://ode.rsl.wustl.edu/mars/productPageAtlas.aspx?product_id=K01_053927_2040_XI_24N151W&product_idGeo=27141575 (accessed on 27 January 2018) |

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
