# Peer review of "Comparison of Topographic Roughness of Layered Deposits on Mars"

_remotesensing, doi:10.3390/rs15092272_

Round 1
Reviewer 1 Report
In this paper, the authors design a multiwavelet algorithm to characterize the two-dimensional topography information for different forms of Martian layered deposits. And the similarity of roughness at multiple scales between inner facies of layered ejecta deposits and long-runout landslides indicate similar emplacement mechanisms, and outside facies of layered ejecta deposits share more similarities with LDAs. However, on the whole, the reader feels that the author uses only one simple parameter (skewness with only three points) to discuss the formation and evolution of the Martian craters’ ejecta, especially with the participation of the water/ice. In fact, the shape of the impact crater ejecta materials is determined by many geological processes and factors. For example, the formation age of the impact crater; the initial stratum thickness, composition, strength, and slope; the physical properties of the impactor; subsequent weathering effect, etc. Therefore, it may be necessary for the authors to further prove that IV3R and multi-scale skewness are very, very critical parameters in studying the morphology of the ejecta from the Martian impact craters.
In addition, the authors need to answer some questions:
Title. The title does not quite match the content of the manuscript. Should “layered deposits” be limited to the layered deposits of the Martian craters?
Figure.2. What are the principles for the authors to choose the long runout landslides and LDAs? Why choose only part of it instead of the whole? In addition, there are only 2 long runout landslides and 3 LDAs in the manuscript. Can such a few examples represent all the characteristics of the long runout landslides and LDAs?
2.2. Method. Lines 131-136. The authors should add more sizes of calculation windows to show more details, instead of just discussing similarities and trends with only three points (60, 132 and 252m).
Figure.5. The authors should provide more details on the selection of LEDs. Why only choose these four areas (L1-L4), and why not choose the ejecta with many cracks on the left of the crater? Moreover, the area between L3 and L4 is significantly rougher than the surroundings, so it is difficult to have an obvious correlation between the roughness of LEDs and the distance from the center of the crater.
Lines 222-224. Can authors reach this conclusion only by relying on two long- runout landslides and three LDAs in this paper? Should more examples be used and compared with previous roughness methods to reflect the progressiveness of the methods and parameters proposed by the authors in this paper?
Lines 276-278. Is this conclusion derived from previous work or from the author's own work?
Lines 279-280. The readers did not see obvious similar skewness values, because the value of 5b is much greater than that of 4a.
Figure. 6. The same question that what are the criteria for selecting LEDs regions? Why didn't the authors choose areas that looked rougher/flatter, such as the ejecta below and to the right of the crater.
In addition, readers would like to see a wider range and more detailed roughness. Therefore, the authors should select the ejecta material of the entire impact crater to apply to the IV3R method and obtain the roughness map of the entire crater ejecta (like Figure. 3h), instead of only providing the skewness values including three points.
Lines 342-343. Please explain this sentence in detail, as the reader has not seen very significant similarities, both in numerical values and in trends.
Reviewer 2 Report
This paper could potentially make a valuable contribution to planetary science. It provides a sophisticated way to analyze layered ejecta blankets from impact craters on Mars, and shows how that analysis can be used to construct plausible conclusions about morphogenetic processes of impact crater formation. Such an analysis would be particularly helpful in suggesting information about the subsurface water content on Mars and similar planetary bodies.
Skewness is clearly a critical factor in this analysis, but its meaning is not clear. Does a more highly skewed distribution indicate a greater degree of roughness? If so this should be stated explicitly, perhaps near line 165.
In like manner, the significance of azimuth is not explained. A brief statement about why measurements at different azimuths is important would be helpful.
Table 1: Shouldn’t the heading of the rightmost column be “Location”? Also, why are there two images given for List #3 while only one is given for all the others?
Fig. 2: Images (d) and (e) are so bright that the topography doesn’t show very well. Is it possible to darken the images somewhat to improve contrast?
Unfortunately, the value of this paper is obscured by significant errors in English composition. The mistakes in usage of articles (presence, absence, or inappropriateness of a, an, or the) and incorrect distinction between singular and plural nouns are too numerous to highlight individually. The authors are strongly urged to have their revised manuscript, if resubmitted, edited in detail by a native or fluent English speaker. The following are further examples that need to be corrected on revision:
Line 14: implications to for subsurface
Line 36: This line doesn’t make sense as written. Did authors mean “. . . grooves and ridges, appearance without the presence of secondary craters” ?
Lines 36-37: morphologies
Line 60: morphology are is similar with part to some features of LEDs, such as the. . .
Line 85: capable to of encompassing topography
Line 117: “DEM” must be defined the first time it is used.
Line 124: root mean square (RMS) slope [17], and median different difference slope (MDS)
Line 127: Is “experientially” really the word the authors intend?
Line 135: wavelet analysis are differed by
Line 144: calculation for to remove the
Line 166: and cannot be applied as a
Lines 204-6: The sentence beginning at line 204 doesn’t make sense. Please recheck your word choice and syntax.
Line 213: . . . highlighted . . .
Line 214: values of these for skewness
Line 239: The base image is derived from [?] seamless mosaic CTX images
Line 242: multi-scales multiple scales
Line 247: deposition, and is characterized
Line 251: the outside outer facies (L3) and the most outside outermost facies (L4)
Line 258: morphology are is characterized
Line 260: An subject ongoing
Line 272-4: This sentence doesn’t make sense. It should be rewritten to make clear what the authors intend
Line 278: process of as long-runout
Line 287: consistent with the fact that L3 shows the rougher topography
Line 289: various processes related to
Line 297-9: This is a long complicated sentence that is difficult to understand. It should be broken into two sentences and reworded for clarity.
Line 302: ferred to as partial
Line 320: focused on the northernmost of the LEDs
Line 328-330: This sentence is too hard to understand. It should be rewritten to improve clarity.
Line 334: Compared By comparing the various
Line 359: skewness values are comparable to
Line 377: to those resulted resulting from landslides
Line 402: measure like 2-norm for to
Line 410: considered that is more suitable for characterize characterizing two-
Reviewer 3 Report
Firstly, I wish authors have given the such a good effort to map and characterize the two-dimensional topography information for different forms of Martian layered deposits by using multiwavelet algorithm. The author’s results show that the similarity of roughness at multiple scales between inner facies of layered ejecta deposits and long-runout landslides indicate similar emplacement mechanisms, and outside facies of layered ejecta deposits share more similarities with lobate debris aprons.
In this research paper, I have few general comments and suggestions which will be useful to update the paper.
Minor comments:
1. Language: the following corrections should be undertaken by authors to fix syntax and style, eliminate typos, and resolve several punctuation problems in the paper.
2. Section 1- In the introduction section, the authors have used the majority of the references were older or previous ones, therefore I advise them to use more recent ones to enhance the paper quality.
3. All figures should be mentioned with exact location name or lati & long location in the fig and text.
4. The conclusion part seems not upright, it has to be developed.
The manuscript can be accepted after the minor corrections.
Round 2
Reviewer 1 Report
The authors responded in detail to the questions and made modifications to the content of the article. I think this article is worth publishing.
Reviewer 2 Report
The authors have done a good job in revising their manuscript to the point where it can now have the positive impact that it deserves. Only a few minor compositional corrections that are needed, which can be handled at the copyreading and proofing stage.